# Maintenance of Flap Endonucleases for Long-Patch Base Excision DNA Repair in Mouse Muscle and Neuronal Cells Differentiated In Vitro

**DOI:** 10.3390/ijms241612715

**Published:** 2023-08-12

**Authors:** Rachel A. Caston, Paola Fortini, Kevin Chen, Jack Bauer, Eugenia Dogliotti, Y. Whitney Yin, Bruce Demple

**Affiliations:** 1Department of Pharmacological Sciences, Renaissance School of Medicine, Stony Brook University, Stony Brook, NY 11794, USA; 2Department of Environment and Health, Istituto Superiore di Sanità, Viale Regina Elena 299, 00161 Rome, Italy; paola.fortini@iss.it (P.F.);; 3Department of Pharmacology and Toxicology, Department of Biochemistry and Molecular Biology, University of Texas Medical Branch, Galveston, TX 77555, USA; 4Department of Radiation Oncology, Renaissance School of Medicine, Stony Brook University, Stony Brook, NY 11794, USA

**Keywords:** oxidative DNA damage, mitochondrial DNA, cell proliferation, mitochondrial DNA endonucleases, alkylating DNA damage

## Abstract

After cellular differentiation, nuclear DNA is no longer replicated, and many of the associated proteins are downregulated accordingly. These include the structure-specific endonucleases Fen1 and DNA2, which are implicated in repairing mitochondrial DNA (mtDNA). Two more such endonucleases, named MGME1 and ExoG, have been discovered in mitochondria. This category of nuclease is required for so-called “long-patch” (multinucleotide) base excision DNA repair (BER), which is necessary to process certain oxidative lesions, prompting the question of how differentiation affects the availability and use of these enzymes in mitochondria. In this study, we demonstrate that Fen1 and DNA2 are indeed strongly downregulated after differentiation of neuronal precursors (Cath.a-differentiated cells) or mouse myotubes, while the expression levels of MGME1 and ExoG showed minimal changes. The total flap excision activity in mitochondrial extracts of these cells was moderately decreased upon differentiation, with MGME1 as the predominant flap endonuclease and ExoG playing a lesser role. Unexpectedly, both differentiated cell types appeared to accumulate less oxidative or alkylation damage in mtDNA than did their proliferating progenitors. Finally, the overall *rate* of mtDNA repair was not significantly different between proliferating and differentiated cells. Taken together, these results indicate that neuronal cells maintain mtDNA repair upon differentiation, evidently relying on mitochondria-specific enzymes for long-patch BER.

## 1. Introduction

As cells differentiate, gene and protein expression changes globally. DNA replication proteins are downregulated in most differentiated cell types as DNA replication in the nucleus ceases [1]. The reduced expression of DNA replication proteins also affects some DNA repair pathways in differentiated cells, which can compromise the processing of oxidative or alkylation damage in nuclear DNA [2,3]. By contrast, mitochondria must maintain and replicate their genomes in order to function and produce new organelles [4]. It is unclear how these processes are affected by cellular differentiation because certain nuclear DNA replication and repair proteins have been implicated in mitochondrial DNA repair [5,6,7]. However, cells that generally do not regenerate, such as most neurons and some types of muscle cells, have to maintain and replicate mitochondria for decades [8]. Thus, defects in mitochondrial DNA repair are associated with neurodegenerative disorders, including Alzheimer’s, Huntington’s, and Parkinson’s diseases [9].

The repair of most oxidative lesions in DNA occurs via base excision DNA repair (BER), which exists in a more limited form in mitochondria compared with nuclei [7]. At the DNA resynthesis step, BER branches into two pathways [10,11,12,13]: a single-nucleotide pathway, which replaces only the damaged nucleotide, and a “long-patch” (multinucleotide) BER (LP-BER) pathway, which is needed for some oxidative lesions [14,15]. LP-BER in mitochondria may involve nuclear replication enzymes [16,17,18,19]. Particularly important for LP-BER are structure-specific endonucleases that remove the displaced oligonucleotide “flap” generated by DNA polymerase when two or more nucleotides are added [5,15].

It is unclear which flap-excising endonuclease(s) support mitochondrial LP-BER. Initially, the nuclear enzymes Fen1 and DNA2 were implicated in this role [17,18,19]. Two other, mitochondria-specific enzymes, ExoG and MGME1, have been suspected to perform flap excision during LP-BER, with the latter also functioning in mitochondrial DNA replication [20,21,22].

Here, we address the availability of flap endonuclease(s) for mitochondrial LP-BER in proliferating and differentiated cells. We implemented two different systems—CAD (Cath.a-Differentiated) mouse neuroblastoma cells [23] and murine satellite skeletal muscle cells [3]—each of which can undergo controlled differentiation in culture. We assessed the expression and activity of specific flap endonucleases, measured the LP-BER activity of whole-cell and mitochondrial protein extracts, and determined the overall repair capacity of the cells. Our results suggest that MGME1 is the predominant flap endonuclease of mitochondria both before and after differentiation.

## 2. Results

### 2.1. Localization and Activity of Flap Endonucleases upon Differentiation

Cellular differentiation in mouse myotubes led to drastically lower levels of the BER proteins DNA ligase I, DNA ligase IIIα, X-ray repair cross-complementing protein 1 (XRCC1), DNA polymerase ε, proliferating cell nuclear antigen (PCNA), and Fen1 protein, while the activities of Ape1 and DNA polymerase β were maintained [3]. Similarly, Ape1 levels in differentiated SH-SY5Y human neuroblastoma cells were only slightly decreased [24] or even modestly *increased* [25,26].

To investigate whether such changes are recapitulated during the differentiation of CAD cells, whole-cell extracts were probed for the expression of several BER-related proteins (Figure 1). After CAD cell differentiation (Appendix A), DNA2 and Fen1 levels were both lowered to <20% of the expression seen in their proliferating progenitors. The expression of both MGME1 and ExoG was less affected following differentiation; these proteins were still expressed at, respectively, ~70% and ~60% of the levels found for proliferating cells, while the level of the Ape1 abasic endonuclease was maintained at ~90% of the level found in cycling cells (Figure 1).

Since Fen1 is present in both the nucleus and mitochondria and is known to function in the nuclear LP-BER pathway, we sought to examine Fen1 expression in CAD cell mitochondria specifically. Despite being downregulated at the whole-cell level, mitochondrial Fen1 was maintained after differentiation and may even have been slightly increased (Figure 2). Immunofluorescence experiments showed that Fen1 colocalized with MitoTracker Red at roughly equal levels in both proliferating and differentiated CAD cells (Appendix A).

To assess total flap excision activity, we assayed CAD whole-cell extracts using a 3′-TAMRA-labeled DNA substrate (Figure 3A). This substrate mimics the flap excision step of the LP-BER pathway (as well as Okazaki fragment intermediates). The 3′-labeled flap substrate showed a clearly lower level (~2-fold) of total flap excision activity in the differentiated cell extracts compared with that in proliferating CAD cells (Figure 3B; quantification in Figure 3C). Initial experiments had been with a substrate containing a 5′-TAMRA label, which may have interfered with the activity of one or more flap nucleases. Nonetheless, those results (Appendix A) were consistent with a somewhat lower level of total activity following differentiation. 

To address the flap excision activity in the mitochondria specifically, mitochondrial extracts were made from dividing or differentiated CAD cells. For this set of experiments, employing the 3′-labeled substrate, we compared the excision products generated by mitochondrial extracts to those produced by the purified candidate flap-excision activities, Fen1, MGME1, and ExoG. The results (Figure 4) indicated that the proliferating and differentiated mitochondrial extracts generated one major product with the same length as that produced by purified MGME1, and similar amounts of a larger (electrophoretically slower) product that co-migrated with only one of the several products observed with ExoG (Figure 4). Note that purified ExoG generated multiple products, but only at the highest level of the enzyme tested (Appendix A). Similar results were observed in three full-scale (biological replicate) experiments.

### 2.2. General Repair Capacity of CAD Cells

Proliferating and differentiated CAD cells were investigated for their mtDNA repair capacity. For these experiments, the cells were treated with either hydrogen peroxide (H_2_O_2_) or MMS, and we first assessed the cell sensitivity to the agents using two different survival assays. As frequently observed for cycling cells, there was substantially greater sensitivity to both H_2_O_2_ and MMS than was observed for terminally differentiated cells (Appendix A). This result was consistent with the greater accumulation of lethal lesions when nuclear DNA is actively being replicated. Additionally, we noted that the quantitative difference between proliferating and differentiated cells was greater for oxidative damage (H_2_O_2_) than it was for the alkylating agent MMS (Appendix A).

The accumulation of mitochondrial DNA damage in cells treated with H_2_O_2_ or MMS was assessed using a PCR-based assay, in which *lower* yields of the amplification product indicate *greater* levels of damage [28]. This study was performed for both proliferating and differentiated CAD cells and for mouse muscle cells differentiated in vitro [3] (Appendix A). Muscle cells demonstrate a similar pattern of survival to that found in CAD cells, with the differentiated cells surviving at higher concentrations of MMS [2]. For the CAD cells, the amount of mtDNA damage was similar for proliferating and differentiated cells after treatment with H_2_O_2_, perhaps slightly lower for the latter (Figure 5A). Following MMS treatment, the mtDNA of differentiated CAD cells exhibited modestly less damage than did the mtDNA of proliferating CAD cells, about twofold at the lower concentrations (Figure 5C). For the muscle cells treated with H_2_O_2_, the mtDNA of the differentiated cells had two- to threefold less damage than did the mtDNA of proliferating cells (Figure 5B). However, the proliferating and differentiated muscle cells accumulated similar levels of damage from the 1 mM MMS treatment, but after treatment with ≥2 mM MMS, the differentiated mtDNA had lower levels of damage than did the mtDNA of proliferating cells, although with noisy determinations (Figure 5D).

## 3. Discussion

The results presented here provide an updated picture of mtDNA repair in differentiated mouse cells. In CAD cells, the overall DNA repair capacity of mitochondrial extracts for lesions processed by LP-BER was comparable to that of whole-cell extracts. This conclusion is consistent with the continued expression of the mitochondrial endonuclease MGME1 in differentiated CAD cells, and that the other mitochondria-localized enzyme tested, ExoG, was only ~50% lower in differentiated than in proliferating CAD cells. Although total Fen1 was sharply downregulated with differentiation as expected, the amount of that protein associated with mitochondria appeared unchanged or even moderately increased. By these criteria, all of these enzymes are candidates to support flap excision during LP-BER in mitochondria.

Careful analysis of the flap excision products (Figure 4) generated by mitochondrial extracts did not show a measurable contribution by Fen1, at least for the substrates used in our experiments. By contrast, the data point to MGME1 as the predominant flap excision enzyme, with ExoG having possible secondary activity. Genetic studies have assigned important roles to MGME1 in mtDNA replication [29,30,31]. With a role for MGME1 in LP-BER, the mitochondrial enzyme would parallel the dual DNA replication and repair roles of Fen1 in the nucleus [32], except that MGME1 levels were not reduced upon differentiation (Figure 1). It is possible that the longer product observed in the mitochondrial extracts (Figure 4) was due to ExoG. However, it remains unclear why additional, shorter ExoG products were not also observed. A role for ExoG cannot be ruled out, but this needs further investigation.

In contrast to mitochondria, DNA flap excision activity in whole-cell extracts was diminished in differentiated cells compared with their cycling precursors, as expected [3,24]. We note that flap excision may not be the rate-limiting step in BER; prior studies have indicated that the levels of DNA ligase IIIα and its essential partner Xrcc1 (associated with single-nucleotide BER) and of DNA ligase I (implicated in LP-BER) are downregulated tenfold or more in both differentiated muscle cells [3] and neuronal cells [24].

When challenged with H_2_O_2_ or MMS, differentiated neuronal and muscle cells both accumulated less mtDNA damage than did proliferating cells. This result could imply that mtDNA repair in differentiated cells is faster than it is in proliferating cells, and it seems unlikely that the mtDNA of differentiated cells was somehow protected from all kinds of DNA damage. However, mitochondria have multiple genomes per organelle, and heavily damaged mitochondrial genomes can be degraded rather than repaired genomes [33,34], which might account for the lower level of damaged mtDNA observed. Another possibility is rapid mitophagy, which was recently observed for mitochondria with DNA containing unrepaired 8-oxoguanine residues [35].

Overall, our results show that at least two cell types maintain the key enzymes necessary to complete LP-BER following differentiation. The somewhat reduced flap excision activity in extracts of differentiated cell mitochondria may indicate more dependence on short-patch BER than on LP-BER for DNA lesions such as uracil. However, there are lesions that require LP-BER, notably abasic sites resulting from 1′-oxidation of deoxyribose [14,15], so the continued function of this pathway would remain important, even in non-cycling cells.

## 4. Materials and Methods

### 4.1. Cell Culture

Proliferating CAD cells were grown in Dulbecco’s Modified Eagle’s Medium mixed in equal proportion with Ham’s F12 medium (DMEM/F12) (HyClone Inc., Logan, UT, USA, #SH30271.01), supplemented with 10% fetal bovine serum (Corning Life Sciences, Inc., Tewksbury, MA, USA, #35-010-CV) and 1% penicillin/streptomycin mixture (Gibco Inc., Billings, MT, USA, #10378016). Differentiation was induced by growing CAD cells in serum-free DMEM/F12 supplemented with 20 µg/mL transferrin and 50 ng/mL sodium selenite. Differentiated CAD cells were grown on plates coated with 20 mg/mL poly-L-lysine. CAD cells were fully differentiated after 5 days [23], and they were always used in the 7 days after terminal differentiation. The formation of dendrites in the differentiated cells is shown in Appendix A).

The isolation of murine satellite muscle cells and their in vitro differentiation into myotubes were performed as previously described [2]. Markers demonstrating the effectiveness of the differentiation procedure are shown in Appendix A.

### 4.2. Quantitative Polymerase Chain Reaction (qPCR) Assay

For H_2_O_2_ or methyl methane sulfonate (MMS) treatments, cells were incubated with the agent for 30 min in a serum-free medium, which was replaced with a serum-supplemented medium after treatment. DNA was extracted using the Qiagen 20/G DNA extraction kit. DNA quantification was performed using the Picogreen reagent. The PCR assay was performed as previously described [26], with the exception of using KAPA LongRange HotStart DNA polymerase, and the following primers were used for mouse mitochondrial DNA:
Long PCR, sense strand: 5′-CCATTCTAATCGCCATAGCCTTCCLong PCR, antisense strand: 5′-GAGGACTGGAATGCTGGTTGGTGGShort PCR, sense strand: 5′-CCCAGCTACTACCATCATTCAAGTShort PCR, antisense strand: 5′-GATGGTTTGGGAGATTGGTGGATG

The short PCR reaction was the control for total DNA content. The long PCR reaction measured the amount of intact DNA. The ratio of these two values indicated the level of DNA damage, such that the lower the value, the more DNA damage was present.

### 4.3. Western Blotting

Cells were lysed using a buffer with sodium dodecyl sulfate (SDS) (Cell Signaling Technology, Danvers, MA, USA, #9803; following the supplier’s protocol), followed by centrifugation at 12,000× *g* for 10 min, and the supernatants were recovered as whole-cell extracts. Protein content was measured using the Bradford reagent. Protein extract samples were loaded on 12% polyacrylamide gels (Invitrogen #NP0343BOX, supplied by Fisher Scientific Co., Boston, MA, USA) and electrophoresed for 2 h at 14.5 V/cm. Wet electroblotting transfer was performed for 3 h on ice onto a polyvinylidene difluoride (PVDF) membrane. The primary antibodies included anti-Ape1 (Novus Inc. Guaynabo, PR, USA, #NB100-101; used at 1:1000), anti-DNA2 (Abcam Inc., Waltham, MA, USA, #ab96488; used at 1:500), anti-ExoG (Abcam #ab77736; used at 1:500), anti-Fen1 (Novus NB100-320; used at 1:700), anti-NDUFA9 (specific for a Complex I protein; Abcam ab14713; used at 1:1000), and anti-TFAM (Cell Signaling 7495; used at 1:1000). The secondary antibodies (both used at 1:10,000) were goat anti-rabbit IgG (Licor Inc., Omaha, NB, USA, #925-68021) and goat anti-mouse IgG (Licor 926-32210). The developed blots were placed in distilled water, scanned on a Li-COR Odyssey (Licor Inc. Omaha, NB, USA), and quantified using Image Studio software (version 5.5.4), with *n* ≥ 3.

### 4.4. Mitochondrial Extraction

CAD cells (1.7 × 10^8^) were centrifuged at 900× *g* in a Sorvall RT Legend centrifuge at 4 °C, and the cell pellets were gently resuspended in chilled hypotonic buffer (20 mM HEPES-KOH, pH 7.4, 5 mM MgCl_2_, 5 mM KCl, 1 mM dithiothreitol (DTT)). The centrifugation and resuspension were repeated once more, and the suspension was incubated on ice for 10 min. The cells were homogenized with 20 strokes of a Dounce homogenizer with the B (tight) pestle, or until ≥90% of cells were broken when viewed under a microscope. The homogenates were then mixed with mannitol/sucrose/HEPES buffer (20 mM HEPES-KOH, pH 7.4, 4 mM ethylenediaminetetraacetic acid (EDTA), 2 mM ethylene glycol-bis(β-aminoethyl ether)-N,N,N′,N′-tetraacetic acid (EGTA), 5 mM DTT, 0.42 M mannitol, 0.14 M sucrose), adding 6 mL of this lysis buffer per 9 mL of homogenate. The mixture was centrifuged twice for 5 min at 9000× *g* in a Sorvall Legend RT to remove nuclei. The supernatant was layered onto 0.8 M sucrose in a centrifuge tube and centrifuged at 10,000 rpm for 10 min in an HB6 rotor (16,000× *g*). The resulting pellet was resuspended in 1 mL mannitol/sucrose/HEPES buffer supplemented with 50 µg/µL bovine serum albumin (BSA). A sucrose step gradient was made using 18 mL 0.8 M sucrose and 6 mL 1.5 M sucrose. The solution containing the mitochondria was layered onto the sucrose and centrifuged at 23,000 rpm for 30 min in an SW32 rotor (40,000× *g*). The band of mitochondria at the 0.8/1.5 M sucrose interface was recovered and diluted into 4 volumes of 20 mM Hepes-KOH (pH 7.4), 4 mM EDTA, 2 mM EGTA, 5 mM dithiothreitol, 0.42 M mannitol, and 0.14 M sucrose. The suspension was then centrifuged at 14,000 rpm for 10 min. The mitochondrial pellet was treated for 3 min with 0.5 mg/mL proteinase K at room temperature. The reaction was stopped with 2 mM phenylmethylsulfonyl fluoride and Roche cOmplete protease inhibitor cocktail, and the mixture was centrifuged at 14,000 rpm for 10 min. The mitochondria in the resulting pellets were then lysed by adding 50 µL of radioimmunoprecipitation assay (RIPA) buffer [36], and the protein concentrations were measured using the Bradford reagent.

Alternatively, for smaller-scale extracts, the Qiagen Qproteome Mitochondria Isolation Kit (#37612) was used according to the supplier’s instructions.

### 4.5. MTT [3-(4, 5-Dimethylthiazolyl-2)-2, 5-Diphenyltetrazolium bromide)] Cell Viability Assay

Cells were plated (20,000 per well) into 96-well plates and incubated overnight. The medium was removed, and the indicated amount of H_2_O_2_ or MMS was then added to the cells in serum-free media. After 30 min, the medium was replaced with a fully supplemented medium, and the cells were allowed to recover for 18 h. The medium was then removed and replaced with a phenol-free medium containing 1.2 mM MTT reagent. After a 2 h incubation, the medium was removed, and the cells were lysed with 0.04% HCl in isopropanol. A plate reader was used to record absorbance at 595 nm and 660 nm; the 660 nm reading accounted for cellular debris and was subtracted from the 595 nm reading, which measured MTT.

### 4.6. Immunofluorescence

Cells were grown in 12-well plates on top of coverslips in the wells and incubated with MitoTracker Red diluted to 0.3 µg/mL in FBS-free media for 30 min. After removing the MitoTracker solution, the samples were incubated for 2 h at room temperature in blocking buffer (0.3% BSA, 0.2% Triton X-100) in PBS (137 mM NaCl, 2.7 mM KCl, 8 mM Na_2_HPO_4_, and 2 mM KH_2_PO_4_), and anti-Fen1 (Abcam #70815, diluted 1:700 in blocking buffer) was added and incubated overnight at 4 °C. The cells were washed twice in PBS, then incubated in FITC-labeled anti-rabbit IgG (1:50) for 1 h. The samples were washed twice with PBS, and the coverslips were mounted on slides using Fluormount-G (SouthernBiotech, Birmingham, AL, USA, #0100-01). Imaging was performed on a Zeiss Confocal LSM 510 microscope.

### 4.7. Substrates

Flap Substrate
FLAPGT13: 5′-GATGTCAAGCAGTCCTAACTTTTTTTTTTTTTTTTGAGGCAGAGTCC-TAMRAFLAP3B1: 5′-CACGTTGACTACCGTCGFBR1G: 5′-GGACTCTGCCTCAAGACGGTAGTCAACGTG

### 4.8. Flap Excision Assays of Whole-Cell and Mitochondrial Extracts

Cell pellets or mitochondrial pellets were resuspended in Buffer I (10 mM Tris-HCl, pH 7.8, 200 mM KCl) at a volume approximately equal to the pellet size. Then, an equal volume of Buffer II (600 mM KCl, 2 mM EDTA, 40% (*v/v*) glycerol, 0.2% NP-40, 2 mM DTT, 0.5 mM phenylmethylsulfonyl fluoride, Roche cOmplete protease inhibitor cocktail) was added; the mixture incubated for 1.5 h at 4 °C and then centrifuged at 10,000× *g* at 4 °C for 10 min. Protein was estimated using the Bradford assay. Extract samples of 0.1, 0.5, or 1 µg were mixed with 1 pmol of the flap substrate in reaction buffer (50 mM Tris-HCl (pH 8.0), 30 mM NaCl, 2 mM DTT, 0.1 mg/mL BSA, 2 mM MgCl_2_, 5% glycerol, 10 mM ATP) to a total volume of 20 µL. Fen1 was isolated in the Demple lab using a vector supplied by Prof. Robert Bambara (University of Rochester) The reaction was incubated for 15 min at 37 °C, terminated by the addition of an equal volume of formamide, and resolved by electrophoresis on 16% polyacrylamide gels containing 25% (*v/v*) formamide and 7 M urea. The resolved bands were quantified via the TAMRA fluorescence.

## Figures and Tables

**Figure 1 ijms-24-12715-f001:**
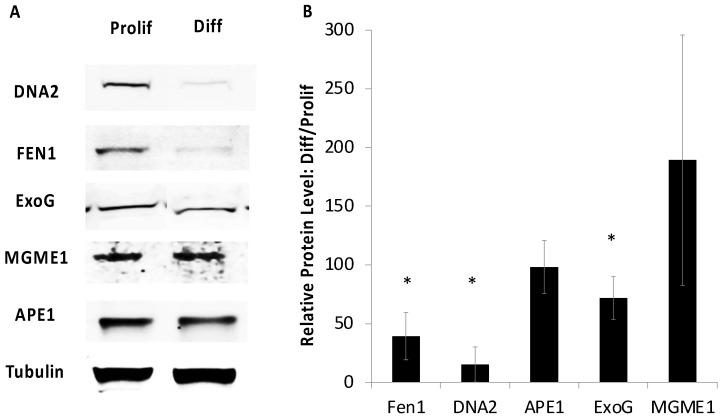
Expression of nucleases upon differentiation. (**A**) Representative Western blots from whole-cell extracts from proliferating (Prolif) and terminally differentiated (Diff) CAD cells. (**B**) Quantification. The signals were normalized to that for tubulin as a loading control, with the Prolif levels set to 100%. * denotes *p* < 0.05 for Diff compared with Prolif., determined using Student’s *t*-test (*n* = 3).

**Figure 2 ijms-24-12715-f002:**
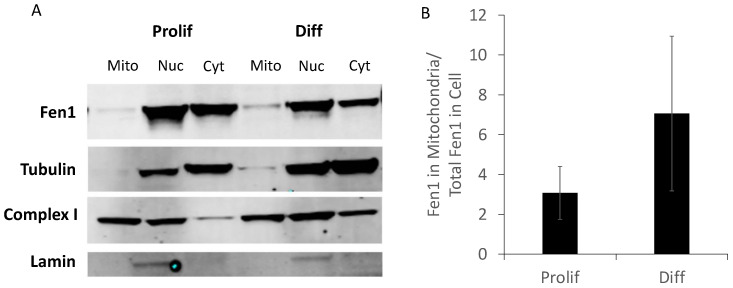
Expression of Fen1 in the mitochondria of differentiated cells. (**A**) Representative Western blots for mitochondrial cell extracts of Prolif and Diff cells. (**B**) Fen1 protein signal was normalized to that for Complex 1 as a loading control for mitochondria, with laminin as a marker for the nucleus. Mitochondrial Fen1 levels were divided by whole-cell Fen1 levels to calculate mitochondrial Fen1 per cell. Error bars indicate the standard deviation (*n* = 3).

**Figure 3 ijms-24-12715-f003:**
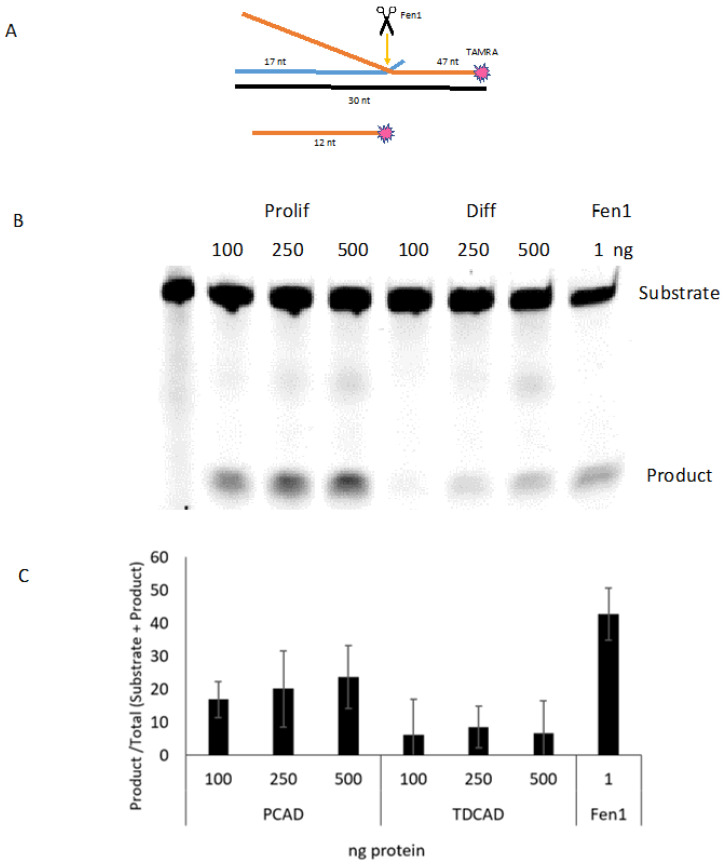
Flap excision activity in proliferating and differentiated CAD cells. (**A**) Substrate for the flap endonuclease assay. The double-flap structure is optimal for Fen1 [27]; the target strand for that enzyme is labeled on its 5′ end the TAMRA fluor. The expected Fen1 product is shown. (**B**) Representative gel for the flap excision assay using whole-cell extracts from Prolif and Diff cells. Increasing amounts of extract were used for 1 pmol of substrate; Fen1 (51 nmol) was used as a positive control. The incubation was performed at 37 °C for 30 min. (**C**) Quantification using Image J (version 2.0.0-rc-68/1.52i) (*n* = 3).

**Figure 4 ijms-24-12715-f004:**
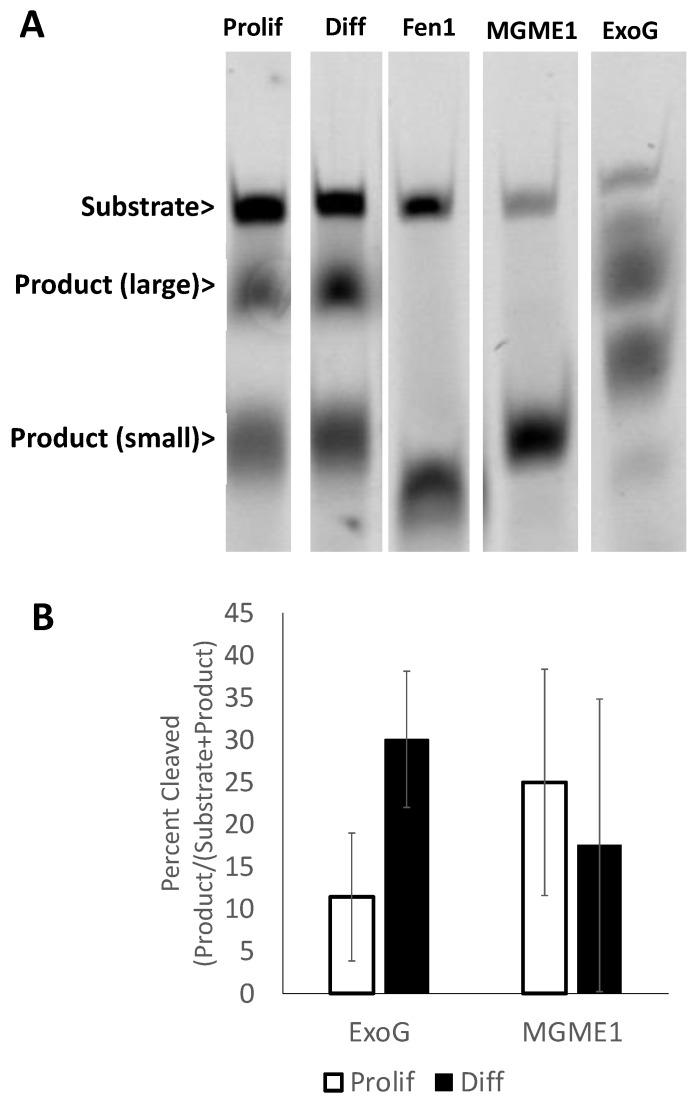
Mitochondrial flap excision activity in proliferating and differentiated CAD cells. (**A**) Lanes from a single representative gel of the flap excision activity using mitochondrial extracts from proliferating (Prolif) and differentiated (Diff) CAD cells. The flap substrate (1 pmol) was incubated at 37 °C for 15 min with 2.7 μg of mitochondrial extract, or 0.5 pmol of Fen1 or MGME1, or 5 pmol of ExoG. The products were resolved by electrophoresis in urea/formamide gels. The original gel is shown in Appendix A. (**B**) ImageJ quantification of cleavage products (*n* = 5), with the slower mobility (large) product taken to indicate ExoG activity and the higher mobility (“small”) taken to indicate MGME1 activity.

**Figure 5 ijms-24-12715-f005:**
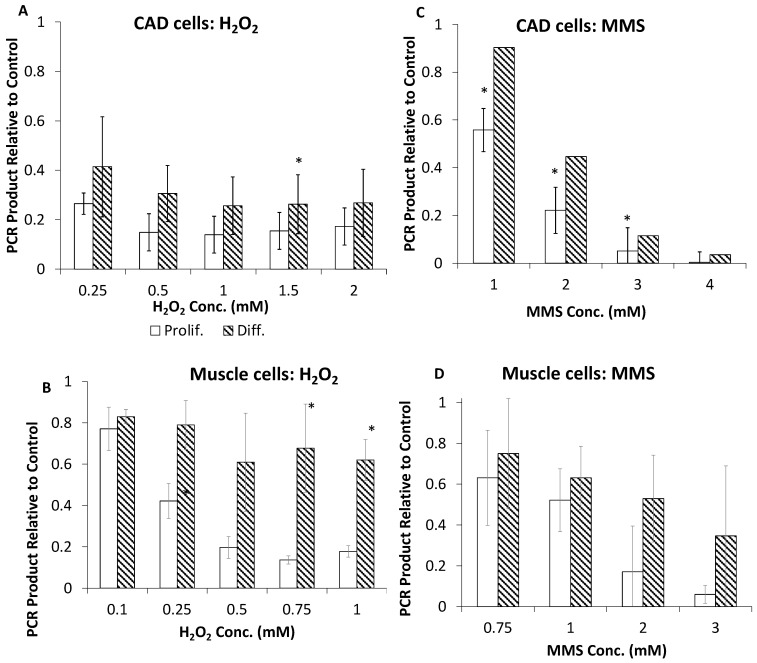
Mitochondrial DNA damage generated by hydrogen peroxide or MMS. (**A**,**C**) CAD cells were treated for 30 min with the indicated concentrations of H_2_O_2_ or MMS. (**B**,**D**) Mouse muscle cells treated as in A and C but for 60 min. Immediately after the treatment, genomic DNA was isolated for PCR analysis. The values for treated cells are reported relative to those for untreated cells, and the result for the controls was set to 1. The error bars indicate standard deviation (*n* = 3), and * denotes *p* < 0.05 between the results for proliferating and differentiated cell mtDNA as measured by Student’s *t* test.

## Data Availability

Data present in and supporting this study are freely available on request to the extent possible. We also freely share the cell lines and vectors in our published work.

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
