# Peer review of "Maintenance of Flap Endonucleases for Long-Patch Base Excision DNA Repair in Mouse Muscle and Neuronal Cells Differentiated In Vitro"

_ijms, 2023, doi:10.3390/ijms241612715_

Round 1

Reviewer 1 Report

In the current manuscript, Caston et al. studied the expression (protein level by WB), sub-cellular localization (IF), and flap excision activity of flap endonucleases, either in total cell lysates or in purified protein form, DNA repair capacity in mitochondria (by qPCR) in proliferating (non-differentiation) and differentiated cells in CAD mouse neuroblastoma cells and murine satellite skeletal muscle cells. They concluded that MGME1 is the predominant flap endonuclease of mitochondria both before and after differentiation.

The work is very interesting; however, the current manuscript was not written properly. Scientifically, I think the current results are very preliminary, more controls should be used and more evidence should be provided to draw meaningful and solid conclusions.

Here are some issues (from major to minor):

1.     Besides CAD cells, other cells, particularly human cells should be included.

2.     Proper controls should be used. For instance, any markers can be used to make sure cells are in Prolif and Diff state? Any markers can be used to make sure cell fractionations are Mito, Nuc, Cyt?

3.     The quantitive data did not match the images shown in Figures (1-4).

4.     Detail figure legends should be given. All figures just have a title, though panels A, B, C… are shown, no explanation given.

5.     Figure 1. B, as my understanding, two columns should be shown for each protein: Prolif and Diff, and the asterisks indicate significant difference between Prolif and Diff. A and B, from my naked eyes, I could see there is difference in APE1 (shown in A), but no difference in APE1 shown in B.

6.     Figure 2. As mentioned earlier, proper controls should be used. Though tubulin, complex I (NDUFA9) and lamin (antibody source was not described) were used here; however, the quality of cellular fractions is a concern, such as complex I. How was the quantitive data (B) calculated? Judged from A, I did not see FEN1 in Diff is more than double of in Prolif, about 7 vs 3. A, besides Fen1, why not check other 3 endonucleases protein level?

7.     Figure 3. B, lane 1 is a negative control? The quantitive data (shown in C) does not match the image (B). For example, I can see the difference among 100, 250 and 500 in Diff group in B, but no difference in C? C, please label Y axis properly.

8.     Figure 4. I understand the results of A, where total cell lysates from Prolif and Diff, and purified proteins (Fen1, MGME1, and ExoG) were used, but I do not understand B, endogenous ExoG and MGME1 proteins were purified from Prolif and Diff? A, here a same substrates and different extracts/purified proteins were used, technically it should be shown on a same image (not separate lanes).

9.     Figure 5 and 6, it is unclear to me how the qPCR works. In M&M long and short PCR primers were listed, which was used as a control (Figure 5 says “relative to control”)? The highest values is 1 in Figure 5, while 2 in Figure 6? Figure 6 says “relative PCR product”, so different measurements were used in these two figures?

10.  Figure 6. I can see 2 lines representing Prolif H2O2 and Diff H2O2, but I do not know which line is Prolif or Diff. Moreover, it was written in line 148: “Cells in both states restored mtDNA integrity at about the same rate, with a half-time of ~1.5 h (Fig. 6)”. Somehow, I did not see this trend, because the solid gray line goes down.

11.  Please describe where the purified FEN1 protein comes from, as well as MGME1 and ExoG. A SDS-PAGE showing these protein is preferred.

12.  Material and Methods section 4.2 qPCR, more info such as internal control and data analysis should be described.

13.  Keywords, might provide more keywords that reflect the topics.

14.  Spell out CAD used in the abstract.

15.  Unify label for Prolif. and Diff. throughout the manuscript.

16.  Please check the language, some expression is not clear.

17.  Just a curiosity: Fen1, DNA2, MGME1 and ExoG work on the same substarate?

Okay, but can be improved for clarity.

Author Response

We appreciate that the reviewer found the study to be “very interesting”.

Below are our responses to the specific points, which the reviewer ordered “from major to minor”.

  1. Besides CAD cells, other cells, particularly human cells should be included.

Please note that, in addition to CAD cells, Figure 5 includes results from mouse skeletal muscle cells.  Of course, we are also curious to know whether the results from our mouse model systems are recapitulated in other systems, including human, and in other cell types.  However, conducting such experiments would be time-consuming, requiring us to set up and verify the new systems; doing the actual experiments with them would then take much additional time, which is beyond the scope of this manuscript.

  1. Proper controls should be used. For instance, any markers can be used to make sure cells are in Prolif and Diff state? Any markers can be used to make sure cell fractionations are Mito, Nuc, Cyt?

The differentiation of CAD cells was routinely verified morphologically; example images are now included in the paper (Fig. S1).  For mouse muscle cells, molecular markers were used to verify differentiation; an immunoblot is provided to demonstrate that (Fig. S6). The sharp down-regulation of Fen1 and DNA2 proteins upon differentiation of both CAD and muscle cells is another molecular indicator of the differentiation. The nuclear and cytosolic fractions were taken during the mitochondrial purification process and were not further purified. Those intermediate fractions are thus expected to show some cross-contamination by other cell fractions. The purpose of including them was as a positive control for the antibodies, and for the comparison of the mitochondrial enrichment in the mitochondrial fraction.  The key thing is that the final mitochondrial fractions are not heavily contaminated with nuclear or cytosolic proteins.

  1. The quantitative data did not match the images shown in Figures (1- 4).

For Figure 1, the order has been corrected such that the image and the bar graph are matched. Figure 3 has been re-quantified to better reflect the image; the prior quantification failed to correct for possible saturation in the most intense bands. Specific concerns about each figure are addressed below.

  1. Detail figure legends should be given. All figures just have a title, though panels A, B, C... are shown, no explanation given.

The figure legends have been edited for clarity.  Other experimental details are in the Methods section.

  1. Figure 1. B, as my understanding, two columns should be shown for each protein: Prolif and Diff, and the asterisks indicate significant difference between Prolif and Diff. A and B, from my naked eyes, I could see there is difference in APE1 (shown in A), but no difference in APE1 shown in B.

What is shown in Fig. 1B is the ratio between differentiation and proliferating cells for each of the indicated proteins.  To avoid possible confusion, the scale for the ratio is now given with 1 indicating an unchanged protein level, so <1 indicates less of the protein in differentiated cells, and >1 indicates an increase.

  1. Figure 2. As mentioned earlier, proper controls should be used. Though tubulin, complex I (NDUFA9) and lamin (antibody source was not described) were used here; however, the quality of cellular fractions is a concern, such as complex I. How was the quantitive data (B) calculated? Judged from A, I did not see FEN1 in Diff is more than double of in Prolif, about 7 vs 3. A, besides Fen1, why not check other 3 endonucleases protein level?

Again, we understand that the explanation was not included. The quantification is explained in the figure legend.  Please note that, because the 3 endonucleases are similar enough in size that western blotting for all 3 required running separate blots, which consumed all of the mitochondrial fraction.  However, each determination was done in triplicate.  The main point in Fig. 2B is that Fen1 was not significantly reduced in mitochondria after differentiation.

  1. Figure 3. B, lane 1 is a negative control? The quantitive data (shown in C) does not match the image (B). For example, I can see the difference among 100, 250 and 500 in Diff group in B, but no difference in C? C, please label Y axis properly.

The first lane contains substrate only, and is thus the enzyme blank, a type of negative control. We have re-quantified the data as noted under point 3 above. The label on the y-axis has been corrected to remove the additional parenthesis.

  1. Figure 4. I understand the results of A, where total cell lysates from Prolif and Diff, and purified proteins (Fen1, MGME1, and ExoG) were used, but I do not understand B, endogenous ExoG and MGME1 proteins were purified from Prolif and Diff? A, here a same substrates and different extracts/purified proteins were used, technically it should be shown on a same image (not separate lanes).

The lanes are from the same gel but, regrettably, there were additional samples in between the lanes of interest that were not needed for this manuscript and which were removed. We can present the entire gel to Reviewer 1 if needed for clarification. Note that MGME1 and ExoG were purified recombinant proteins.  The latter is from the lab of our co-author Prof. Yin; the MGME1 was provided by the lab in Sweden where it was originally identified and isolated (noted in the Acknowledgements); and the Fen1 was isolated in the Demple lab using a vector supplied by Prof. Robert Bambara (University of Rochester).  Of course, that should have been listed in the Methods section, as it now is in section 4.8.  For 4B, the upper and lower bands were quantified. They have been labeled as ExoG and MGME1 because they migrate to the same location as the products of those endonucleases.

  1. Figure 5 and 6, it is unclear to me how the qPCR works. In M&M long and short PCR primers were listed, which was used as a control (Figure 5 says “relative to control”)? The highest values is 1 in Figure 5, while 2 in Figure 6? Figure 6 says “relative PCR product”, so different measurements were used in these two figures?

Figures 5 and 6 were quantified in the same way; for Fig. 5, the y-axis was modified to reflect that, while Fig. 6 has been removed (see below). The short PCR (unlikely to encounter a lesion) was used to measure total DNA, and the long PCR measures the amount of intact DNA, which is diminished by the presence of lesions along the 9-kb template.  The reported values are derived from these values, after setting the ratio for untreated cells = 1; they are thus “relative to control”.  The ratio for the untreated cells is not shown, as it is defined as 1.

  1. Figure 6. I can see 2 lines representing Prolif H2O2 and Diff H2O2, but I do not know which line is Prolif or Diff. Moreover, it was written in line 148: “Cells in both states restored mtDNA integrity at about the same rate, with a half-time of ~1.5 h (Fig. 6)”. Somehow, I did not see this trend, because the solid gray line goes down.

We agree that the variability in Figure 6 makes the data difficult to interpret. Due to the considerable error in the data and potential for confusion from inconsistency in the controls, we removed Figure 6 from this manuscript. Our conclusion, that differentiated cell mitochondria are not deficient in DNA repair, is supported by the results of Figure 5.

  1. Please describe where the purified FEN1 protein comes from, as well as MGME1 and ExoG. A SDS-PAGE showing these protein is preferred.

Please see comment 8 above.

  1. Material and Methods section 4.2 qPCR, more info such as internal control and data analysis should be described.

We have added the following statement:

“The short PCR reaction is the control for total DNA content. The long PCR reaction measures the amount of intact DNA. The ratio of these two values gives indicates the level of DNA damage, such that the lower the value, the more DNA damage is present.”

  1. Keywords, might provide more keywords that reflect the topics.

We have added the keywords: Mitochondrial DNA endonucleases, alkylating DNA damage

  1. Spell out CAD used in the abstract.

Corrected.

  1. Unify label for Prolif. and Diff. throughout the manuscript.

Corrected.

  1. Please check the language, some expression is not clear.

We have reviewed carefully for correctness and readability.

  1. Just a curiosity: Fen1, DNA2, MGME1 and ExoG work on the same substarate?

Yes, these enzymes can work on an overlapping set of substrates, but as shown in Fig. 4, they produce distinct products.

Reviewer 2 Report

Caston et al. tackle the question of how mitochondrial base excision repair (BER) can operate in in differentiated neuronal cells given that the nuclear flap endonucleases are down-regulated, along with most replication-associated enzymes, when DNA is no longer being replicated.  BER will continue to be required for repair of alkylation and oxidative damage to DNA.  By comparing flap endonuclease levels in growing and differentiated cells, they show that DNA2 and Fen1 are indeed much less present in differentiated cells.  However, two mitochondrial flap endonucleases are not down regulated.  

In vitro assay of flap endonuclease activity in whole cells shows that such activity is diminished in differentiated cell extract.  Characterization of the products of flap endonuclease activity show that Fen1 does not produce the products seen in either proliferating or differentiated cell mitochondria.  MGME1 produces a major band.  ExoG produces the smaller band, but also another band that is not seen with mitochondrial preparations.  So, the flap endonucleases of mitochondria are not easily explained by these enzymes.

Quantitative PCR was used to monitor the amount of DNA damage caused by peroxide and by an alkylating agent in growing and stationary cell mitochondria.   Although this technique seems to be very indirect, it is well established in the literature.  Overall, there is more PCR product after treatment in differentiated cells, implying robust repair if one assumes equal susceptibility to DNA damage.  It is noted that interpretation might also be difficult because loss of mitochondrial DNA and mitophagy are not known.  

Overall, this seems to be a sound set of experiments and a well written paper.  The findings constitute a significant contribution to our understanding of DNA repair in the mitochondria of nerve cells that are no longer proliferating. 

Author Response

We are grateful that this reviewer has found our paper worthy of publication.

Round 2

Reviewer 1 Report

Authors have made improvement for the manuscript, mainly on writing; however, the scientific quality can be improved to publishable level.

Author Response

We have done our best to improve the readability and clarity of the paper in response to the points of the academic editor.